# The Cardiomyocyte in Heart Failure with Preserved Ejection Fraction—Victim of Its Environment?

**DOI:** 10.3390/cells11050867

**Published:** 2022-03-02

**Authors:** Angela Rocca, Ruud B. van Heeswijk, Jonas Richiardi, Philippe Meyer, Roger Hullin

**Affiliations:** 1Department of Cardiology, Faculty of Biology and Medicine, Lausanne University Hospital, University of Lausanne, 1011 Lausanne, Switzerland; angela.rocca@chuv.ch; 2Department of Diagnostic and Interventional Radiology, Faculty of Biology and Medicine, Lausanne University Hospital, University of Lausanne, 1011 Lausanne, Switzerland; ruud.van-heeswijk@chuv.ch (R.B.v.H.); jonas.richiardi@chuv.ch (J.R.); 3Cardiology Service, Department of Medical Specialties, Faculty of Science, Geneva University Hospital, University of Geneva, 1205 Geneva, Switzerland; philippe.meyer@hcuge.ch

**Keywords:** heart failure with preserved left ventricular ejection fraction, cardiomyocyte, animal models

## Abstract

Heart failure (HF) with preserved left ventricular ejection fraction (HFpEF) is becoming the predominant form of HF. However, medical therapy that improves cardiovascular outcome in HF patients with almost normal and normal systolic left ventricular function, but diastolic dysfunction is missing. The cause of this unmet need is incomplete understanding of HFpEF pathophysiology, the heterogeneity of the patient population, and poor matching of therapeutic mechanisms and primary pathophysiological processes. Recently, animal models improved understanding of the pathophysiological role of highly prevalent and often concomitantly presenting comorbidity in HFpEF patients. Evidence from these animal models provide first insight into cellular pathophysiology not considered so far in HFpEF disease, promising that improved understanding may provide new therapeutical targets. This review merges observation from animal models and human HFpEF disease with the intention to converge cardiomyocytes pathophysiological aspects and clinical knowledge.

## 1. Introduction

Heart failure (HF) afflicts an estimated 64 million people world-wide [1] and the high hospitalization rate of HF patients, particularly among the elderly [2], accounts for up to 10% of the healthcare costs for cardiovascular disease [3]. Definitions of the societies of cardiology [4] divide the HF population into two large categories as a function of the left ventricular ejection fraction (LVEF): HF with LVEF < 50%, which is further subdivided into HF with mildly reduced LVEF from 40 to 49% (HFmrEF) and HF with reduced LVEF < 40% (HFrEF), while HF patients with a preserved LVEF ≥ 50% (HFpEF) make up the other large category. Today, HFpEF already comprises more than half of the total HF population [5,6], and this proportion is supposed to rise in the near future, since the annual HFpEF incidence has increased by about 1% relative to the incidence of HF with LVEF < 50% in the last years [7,8].

The last decades have seen the successful evolution of medical treatment, which improved survival and decreased morbidity in HFeEF. But until today, understanding of HFpEF remains poor and treatment options that improve prognosis remain limited to HF patients with a LVEF of 50–60%, or even lower, as attested repeatedly in large randomized trials such as the TOPCAT study testing spironolactone, the PARAGON-HF trial comparing LCZ696 with enalapril, and the EMPEROR-Preserved study with empagliflozine in the test group [9,10,11]. The fact that molecules of different pharmacological classes are effective in HF patients with LVEF < 60% but not in HF patients with LVEF ≥ 60% challenges the current categorization of HF patients on the basis of the definition provided by the societies of cardiology [12], suggesting that based on the results the available clinical trial results, HF can be categorized into HF with reduced or normal LVEF. This is in coherence with arguments suggesting that HF is a heterogeneous disease with a dynamic evolution of functional and structural changes, leading to unique disease trajectories and a spectrum of phenotypes with overlapping and distinct characteristics [13].

HFpEF usually does not trace back to a primary injury of the cardiomyocyte as is often the case with HFrEF [14]. Instead, an individual collection of systemic abnormalities such as aging, hypertension, and metabolic stress is possibly at the origin of the disease [7,15]. While this multifaceted cause may explain phenotype heterogeneity in HFpEF [16], it has complicated comprehension of the interplay between these systemic abnormalities and cardiomyocyte structure and function. However, progress is of urgent importance since five-years survival of HFpEF is a dismal 35% after an index HF hospitalization, worse than for most cancers. Furthermore, morbidity is high in HFpEF and the care of these patients will likely present a major societal burden in the future [16,17,18].

Very recently, major pathobiochemical aspects of HFpEF were comprehensively summarized [19]. The goal of this review is therefore to join state-of-the-art understanding of cardiomyocyte pathology in HFpEF disease with current clinical knowledge for a synoptic view on this challenging category of HF disease.

## 2. The Dilemma of HFpEF Definition

Heart failure is a clinical syndrome that consists of cardinal symptoms (breathlessness, ankle swelling, fatigue, and others) accompanied by typical clinical signs (elevated jugular venous pressure, pulmonary crackles, and peripheral edema). Its origin lies with the structural and functional pathology of the heart, which results in increased cardiac filling pressure and inadequate cardiac output at rest and/or during exercise [14]. This definition is valid for HF patients with reduced as well as preserved LVEF [4]. While increased left ventricular end-diastolic filling pressure (LVEDP) results predominantly from systolic dysfunction in HFrEF patients, it is diastolic dysfunction or myocardial stiffness that is mostly the cause of increased LVEDP in HFpEF. Increased filling pressure can be therefore considered as the objective gold standard for diagnosing HFpEF. Since not all HFpEF patients will undergo hemodynamic testing, non-invasive definitions were developed by the different societies of cardiology based on clinical and echocardiographic characteristics of this disease [20]. When tested in a cohort of 461 symptomatic patients with exertional dyspnea and LVEF ≥ 50% but without exercise-limiting pulmonary disease, each definition of the major societies of cardiology identified patients of this cohort at a variable percentage. However, only a small portion of patients was diagnosed with all characteristics cited above. Moreover, only 243 of the patients of this cohort had increased filling pressure at rest in invasive hemodynamic analysis, suggesting that the universally valid physiologic HF definition of increased LVEDP may not apply for the whole clinical entity of HFpEF [20]. Comprehensive hemodynamic analysis therefore helps to refine HFpEF subgrouping, but as a standalone may miss those patients not fulfilling the standard hemodynamic criteria of HF. These can be HFpEF patients of early-stage disease or old patients with age-related cardiac pathology such as hypertension, exercise limitation, but normal LVEDP at rest. However, N-terminal prohormone of brain natriuretic peptide (NT-proBNP) and high sensitivity C-reactive protein (hs-CRP) levels were also considered as biomarkers of HFpEF allowing the characterization of patients. In summary, these findings highlight the phenotype complexity and heterogeneity of current HFpEF definitions, and question whether HFpEF subgrouping on the basis of the existing definitions reliably supports testing of potential therapeutic strategies.

### 2.1. Structural and Functional Changes of the Cardiomyocyte in HFpEF

Independent of this diagnostic dilemma, enormous progress has been made in the comprehension of HFpEF disease in recent years with translational studies that investigated the origin of diastolic dysfunction. A remarkable study in the human investigated left ventricular epicardial anterior wall biopsy specimens obtained from patients with LVEF ≥ 50%, normal wall motion, and end-diastolic volume index < 75 mL/m^2^ undergoing coronary artery bypass grafting [21]. Patients were divided into groups without hypertension (HTN) (control group), with HTN but without HFpEF, and HTN with HFpEF. Compared to controls, patients with HTN but without HFpEF had no change in LVEDP, myocardial passive stiffness, collagen, or titin phosphorylation, but showed an increase of serum biomarkers of inflammation (C-reactive protein, ST2, tissue inhibitor of metalloproteinase-1). However, patients with HTN and HFpEF had high LVEDP, increased left atrial volume, and elevated levels of NT-proBNP or biomarkers of inflammation. Most importantly, overall stiffness was increased with collagen and titin contributing simultaneously based on correlation coefficient analysis. In summary, these changes suggest that HFpEF involves changes in collagen and titin homeostasis and that proinflammatory and profibrotic stimuli may play a role [21].

In fact, the giant cytoskeletal protein titin, which is the largest known protein [22], acts as a bidirectional spring and is responsible for early diastolic recoil and late diastolic distensibility of cardiomyocytes. It exists in two cardiac titin isoforms: the more compliant N2BA and the stiffer N2B isoform [23,24] and the N2BA:N2B ratio defines myofibrillar passive stiffness [25,26,27,28]. Of note, in a murine model of HFpEF there is an increase in the expression of the two isoforms with a relative reduction of the N2BA [29]. However, LV stiffness also depends on the phosphorylation state of titin, and, in HFpEF patients when compared to controls titin phosphorylation on PEVK S11878 (S26) (the proline, glutamate, valine, and lysine domain) is more important, while phosphorylation on N2B S4185(S469) is decreased [30]. The phosphorylation state of titin is modulated by different protein kinases: the cAMP-dependent protein kinase (PKA), the cGMP-dependent protein kinase (PKG), and the protein kinase C (PKC) [31]. Phosphorylation by PKA or PKG render titin more compliant, while protein-kinase-C-dependent phosphorylation decreases compliance [32]. In detail, phosphorylation of N2B (N2Bus segment) reduces cardiomyocyte resting tension (F_passive_), while PKC mediated phosphorylation at the PEVK domain increases stiffness [33,34]. Therefore, a deficit in phosphorylation at the N2Bus-titin site by PKA/PKG or an increased phosphorylation at the PEVK domain by PKC leads to more important F_passive_ and thus stiffness [33,34]. PKG activity is in general decreased in HFpEF, while PKC is elevated, resulting in imbalanced phosphorylation and increased diastolic stiffness [30,35,36,37] (Figure 1). This can explain why sildenafil reduces diastolic stiffness in an old hypertensive dog model and also in HFpEF patients on chronic sildenafil treatment for pulmonary hypertension [23].

PKG also functions as a brake on myocardial hypertrophy and lower PKG activity was furthermore shown to correlate with larger cardiomyocyte diameter [37], which is an acknowledged histological characteristic of HFpEF [24]. In a murine model of transverse aortic restriction introducing acute LV afterload increase, sildenafil, which inhibits cGMP breakdown via phosphodiesterase type 5A (PDE5A), resulted in increased activation of PKG and consequently a decrease of cardiomyocyte hypertrophy and cardiac fibrosis [38] (Figure 1). Moreover, sildenafil treatment reduced LV mass and mass/volume ratio in diabetic cardiomyopathy patients with concentric remodeling [39], emphasizing the important role of the PKG on the one hand, while simultaneously suggesting augmentation of PKG activity as a potential therapeutical target on the other hand. However, 24-week administration of sildenafil for phophodiesterase-5-inhibition did not result in significant improvement of the exercise capacity or clinical status among HFpEF patients when compared with normal controls [40]. However, a post-hoc analysis of the RELAX-HF trial and of two other HFpEF trials [9,41] indicates that all these trials included patient subgroups with different cardiac and non-cardiac comorbidities [42], which may explain the overall neutral study result of these trials.

Homeostasis of cardiac contractile function requires refined regulation of sarcomeric protein phosphorylation such as of the troponin complex of the thin filaments, which are substrates for multiple kinases [43]. In HFrEF, phosphorylation of sarcomeric proteins plays an important role [44,45,46], but whether phosphorylation of sarcomeric proteins plays a role in HFpEF remains uncertain. Nonetheless, hypophosphorylation of the regulatory myofilament proteins cTnI, cMyBPC, cMLC2 goes along with increased calcium sensitivity, suggesting that functional impairment at the sarcomere level may play a role in HFpEF, all the more since cytolsic calcium levels are increased in animal models of HFpEF [30].

### 2.2. The Impact of Age, Hypertension, Obesity, and Diabetes in HFpEF

Earlier animal models investigated aspects of HFpEF disease, in particular LV hypertrophy and diastolic dysfunction either in the TAC mouse model [37] or in the old-dog model with perinephritis induced hypertension [23]. However, these models do not reflect the complexity of HFpEF, where age, diabetes, hypertension, diabetes, and obesity simultaneously impact cardiac function and morphology [47].

In humans, aging is associated with an increased prevalence of LV hypertrophy and diastolic dysfunction, even in the absence of arterial hypertension [48,49]. Old hearts from humans or animals are likewise noteworthy for both enlarged cardiomyocytes and a decreased number of cardiomyocytes due to increased apoptosis and necrosis [43,50]. Similar changes are recapitulated in murine aging models showing increased passive stiffness and decreased active diastolic relaxation properties of the cardiomyocyte [51]. These changes go together with accumulation of mitochondrial protein oxidation as well as increased mitochondrial biosynthesis and mitochondrial DNA mutations as a consequence of increased ROS production [52] (Figure 1). Moreover, compensatory cardiac remodeling with alterations in extracellular matrix composition and an increase of cardiac fibroblast numbers were observed both in the aged human and rodent heart [53]. Altogether, these changes suggest aging as a key determinant of exertional fatigue and/or dyspnea in human and rodent HFpEF. This is supported by the finding that old age clustered 25% of the 6909 HFpEF patients in follow-up by the Swedish Heart Failure (SwedeHF) in a latent class analysis [54].

Systemic hypertension is the single most common comorbidity in HFpEF with a prevalence ranging from 60 to 96% in epidemiological studies, HF registries and large randomized controlled trials [10,55]. The relationship between hypertension and LV hypertrophy is well known since elevated blood pressure in midlife results in LV hypertrophy in later life [56]. Systemic hypertension furthermore induces arterial stiffness, which by itself results in a disproportionate afterload increase, up to the point of ventricular-vascular uncoupling [57]. Animal models such as the aldosterone-infused and unilateral nephrectomized mouse and the angiotensin-II infused mouse reproduced these characteristics. Both murine models develop hypertension, concentric LV hypertrophy, pulmonary congestion, and evidence of diastolic dysfunction and exertion intolerance while maintaining a normal or preserved LVEF. Beyond these typical clinical characteristics repeating human HFpEF, these mouse models reproducing other aspects of human HFpEF such as changes in titin NB2A and NB2B expression and an increase of extracellular matrix quantity [29,30,58]. This concordance suggests that animal models repeating chronic hypertension can serve for further translational studies in HFpEF. Results from epidemiological studies in humans indicate the importance of arterial hypertension since arterial hypertension was a key parameter in three of the five different clusters identified by latent class analysis of the SwedeHF study. In detail, arterial hypertension clustered with non-diabetic participants presenting with atrial fibrillation (30% of all cohort participants), obese patients (15%), and older participants with ischemic heart disease and renal dysfunction (20%) [54].

However, neither the rodent aging models nor the hypertension models contain aspects of the metabolic comorbidity in HFpEF, in particular obesity and diabetes. In fact, large outcome trials and registries reveal that being overweight or obese is a major risk factor of HFpEF [7]. There are multiple mechanisms whereby obesity contributes to HFpEF, since adiposity increases systemic inflammation, insulin resistance, and dyslipidemia, and impairs arterial, skeletal muscle, and physical function [59], all of which are abnormal in HFpEF [60] (Figure 1). In contrast, weight loss after bariatric surgery in the human decreases LV hypertrophy, and LV filling pressures and reduces diastolic dysfunction [61]. Furthermore, exercise capacity improves with caloric restriction and aerobic exercise training in human HFpEF [62]. HFpEF patients with morbid obesity (mean body mass index (BMI) 41 kg/m^2^) behave differently when compared to HFpEF patients with elevated systolic blood pressure and left ventricular hypertrophy. Their hemodynamic burden is less important, the systolic force-Ca^2+^ dependence is reduced by 50%, indicating depressed contractility, and passive stiffening is less important [63,64]. The molecular mechanisms for obesity-associated depression of sarcomere function remain largely unknown, but these findings call conventional views about HFpEF and cardiac contractility into question. Further support for a role of BMI in HFpEF derives from RNA sequencing of right ventricular septal endomyocardial biopsies of HFpEF patients. This study showed minimal overlap with gene expression in HFrEF and healthy donor controls in a principal component analysis. Upregulated genes in human HFpEF were enriched in mitochondrial adenosine triphosphate synthesis/electron transport, while these pathways were downregulated in HFrEF. Body mass index largely accounted for upregulated genes in HFpEF, whereas neither BMI nor other comorbidity was associated with pathways enriched in downregulated genes such as genes engaging endoplasmic reticulum stress, autophagy, and angiogenesis [65] (Figure 1). Yet, information on the effects of obesity alone from animal models is sparse since most animal models also present obesity with either insulin resistance or diabetes [24]. Thus, replication of this association of upregulated gene expression with BMI in an animal model is missing so far.

Diabetes and obesity are two components of the cardiometabolic syndrome, which comprises also arterial hypertension and dyslipidemia and the prevalence of diastolic dysfunction is 12–45% with increasing severity of obesity, 50% in pre-diabetes, and 70% in type 2 diabetes [66]. In particular, diabetes confers a significant additional increase of the relative risk for cardiovascular death and hospitalization in all HFpEF patients [67]. The major pathomechanisms involved with diabetes are cardiac insulin resistance as well as neurohormonal and cytokine imbalance. Insulin resistance results in insufficient energy supply of the cardiomyocyte with subsequent hypophosphorylation resulting in increased cardiomyocyte stiffness and impairment of diastolic function, whereas cytokine imbalance activates the mitogen-activated protein kinase (MAPK) signaling pathway, which again favors myocellular hypertrophy and cardiac fibrosis. [67] (Figure 1).

Most murine models of obesity and diabetes-induced HFpEF face the limitation that the rodent phenotype develops most often fulminant, while the human phenotype develops over years or even a longer time. One of the animal models that repeats slowly developing HFpEF is the db/db leptin-deficient mouse, which develops obesity concomitant with severe hyperglycemia due to type 2 diabetes. In early disease (eight weeks), animals show increase of inflammatory cytokines levels, whereas there is decrease at older age (28 weeks). Furthermore, LV hypertrophy develops only at older age along with enlarged cardiomyocytes, an increase of cardiac fibrosis and capillary rarefication. In addition, this mouse shows abnormal ventriculoarterial coupling due to decreased arterial compliance and increased LV stiffness [68], which repeats a central characteristic of human HFpEF (Figure 2B). Heart rate reduction with ivabradine improved in this murine model vascular stiffness, LV contractility, and diastolic function [69], but this treatment was without effect on cardiovascular outcomes in HFpEF patients [70]. Again, this result may question the applicability of results obtained in rodent studies; however, the result also suggests that the drug was not provided to the phenotype subtype of HFpEF patients with potential benefit from I_f_ channel inhibition.

Another useful model for understanding the HFpEF phenotype is the ZSF1 rat, which was developed by crossing rat strains with two different leptin mutations (fa and fa^cp^), the lean female Zucker diabetic fatty rat (+/fa) and the lean male spontaneously hypertensive HF rat (+/fa^cp^) derived from the obese SHR carrying the corpulent fa^cp^ gene. These animals develop a cardiac phenotype compatible with HFpEF [71] and also show titin NB2B hypophosphorylation and the latter may explain why sildenafil treatment improves the phenotype [72]. However, trials testing sildenafil treatment in HFpEF patients were neutral for the reasons discussed above [9,40,41,42]. In conclusion, the db/db mouse and the ZSF1 rat present a cardiac HFpEF, the obese/metabolic phenotype which is compatible with a cluster of HFpEF patients. The SwedeHF registry composed of patients with obesity, diabetes, and hypertension altogether representing 15% of the all HFpEF patients [54].

However, the db/db mouse leptin deficient mouse and the ZSF1 rat develop functional and structural changes that resemble HFpEF due to genetic pathology and thus do not reproduce the emergence of HFpEF disease on the basis of comorbidity. Therefore, rodent models were developed to study the impact of double or triple comorbidity on cardiovascular function [73,74]. In a two-hit murine model, obesity as induced providing a high fat diet and these animals developed metabolic syndrome with concomitant systemic inflammation in the following. In addition, arterial hypertension was induced by adding L-NAME (N-nitro-L-arginine methyl ester) in the drinking water to drive endothelial-based hypertension. This two-hit model, but not application of either component alone, reproduced pathophysiological changes observed in human HFpEF (Figure 2A) [75], and resulted in increased expression of inducible nitric oxide synthase (iNOS), similar to observations in the ZSF-1 obese rat. Elevated iNOS activity promotes S-nitrosylation of cysteine residues of multiple proteins perturbing their function [76] and including the inositol requiring endoribonuclease (IRE1α) [76]. S-nitrosylation of IRE1α resulted in defective splicing of X-box-binding protein 1 (XBP1 s) culminating in deranged unfolded protein response (UPR), an evolutionary conserved adaptive response capable of mitigating stress in conditions that disrupt protein quality control in the cardiomyocyte (Figure 1). In fact, the UPR is a ubiquitous signal transduction pathway sensing the fidelity of protein folding in the endoplasmic reticulum. It transmits information on the protein folding status to the nucleus and the cytosol and thereby adjusts the protein folding capacity of the cell or, in the event of chronic damage, induces apoptotic cell death. The observation that systemic inflammation and nitrosative stress affect protein homeostasis via suppression of the IRE1α-XBP1 axis of the UPR in the two-hit model [77] and the presence of multiple features of the two-hit models recapitulating HFpEF disease highlight a clinically relevant role of the UPR in this disease. In fact, perturbation of this pathway has been related to other systemic disease such as diabetes and obesity, and therefore disorder of this higher-order regulatory pathways could explain multiple facets of HFpEF disease.

Furthermore, a three-hit strategy combining age, long-term high-fat diet, and deoxycorticosterone pivalate challenge resulted in a mouse model that replicated key hemodynamic features of HFpEF (Figure 2A). In detail, three-month-old mice were fed a high-fat diet, resulting in early-on obesity and impaired glucose tolerance. The high-fat diet was continued for 13 months, and in the last month of this high-fat diet, deoxycorticosterone pivalate was injected intraperitoneally to accentuate hypertension and systemic inflammation. With this three-hit strategy, the mice developed LV hypertrophy, hypertension, impaired endothelium-dependent vasorelaxation, and evidence of increased oxidative stress and inflammation. Furthermore, the animals showed marked collagen deposition and global increase of interstitial fibrosis. Moreover, these mice presented typical hemodynamic characteristics of HFpEF with impaired diastolic relaxation and a reduced end-diastolic LV volume as shown in LV pressure-volume loop analysis. These changes went along with reduced expression of mitochondrial deacetylase sirtuin 3 (Sirt3), resulting in hyperacetylation of many mitochondrial proteins (Figure 1) [73]. Hyperacetylation promoted activation of the ASC protein (apoptosis-associated speck-like protein with a caspase-recruitment domain), which is a key element of the inflammasome connecting the sensor protein NLRP3 (NOD-like receptor proetin-3) to the effector protein caspase-1. The spatial arrangement of these three proteins activates the NLRP3 -dependent inflammasome increasing the secretion of the proinflammatory interleukins IL-1ß and IL-18 in this mouse model and in blood samples of HFpEF patients. Of note, hyperacetylization of ACS favors inflammasome-formation in the mitochondria [73] and the phenotype of the three-hit mouse model was more severe when the experimental protocol was applied to deacetylase Sirt3-KO mice confirming the important role of hyperacetylation. In addition, this observation expands on a previous study showing that increase of hyperacetylation in HF due to increase of the nutrient-based mitochondrial acetyl-CoA pool promotes progression of HF when combined with reduced deacetylase activity [77].

Analysis of the acetylom showed that a large portion of the hyperacetylated proteins are in the mitochondria and a large number of hyperacetylizated proteins belong to the TCA cycle and proteins involved in oxidative phosphorylation and fatty acid oxidation [78]. It is therefore not surprising that hyperacetylation in the HFpEF hearts correlates with a reduced NAD^+^/NADH ratio, impaired mitochondrial function, and cellular depletion of TCA cycle metabolites. Of note, supplementation of nicotinamide riboside normalized the NAD^+^/NADH ratio in the three-hit model, downregulated the acetylization level, improved mitochondrial function and ameliorated the HFpEF phenotype. These observations are in line with other reports suggesting a role of acetyl-CoA, activity of deacetylase such as Sirt3, and acetylases in the pathogenesis of HF [79]. The importance of the NAD^+^/NADH ratio was already suggested by a cardio-specific complex 1 deficiency model where normalization of the NAD^+^ redox balance by nutrient-based supplementation with nicotinamide mononucleotide protected this mouse heart from pathologic hypertrophy and contractile dysfunction induced by chronic pressure overload [80,81]. Deficiency of the TCA cycle due to acetylization was also shown in a mouse with an acetyl-mimetic mutation targeting a succinate dehydrogenase A lysine residue shown, which had been demonstrated to be hyperacetylated in the failing heart. Acetylation of this residue reduces catalytic function and reduces complex 2-driven respiration (Figure 1) resulting in an altered mitochondrial acetyl CoA homeostasis and deranged energy supply driving in heart failure [80,81] (Figure 1). However, the exact underlying mechanisms leading to hyperacetylation and development of HFpEF remain only partially understood. Nonetheless, detailed analysis holds the promise to discover future target of therapeutical intervention in HFpEF.

With the large success of the SGLT2 inhibitor empagliflozin improving prognosis for patients suffering from HF, various pathophysiological concepts have been proposed mostly based on animal data [11]. In the absence of HFpEF animal models testing empagliflozin, artificial neural networks and deep learning AI were used to model the molecular effects of empagliflozin EF. The model predicted that empagliflozin could reverse 59% of the protein alterations found in HFpEF. The effects of empagliflozin in HFpEF appeared to be predominantly mediated by inhibition of NHE1 (Na^+^/H^+^ exchanger 1), with the SGLT2 protein playing a less prominent role. Empagliflozin’s pharmacological action mainly affected cardiomyocyte oxidative stress modulation, and greatly influenced cardiomyocyte stiffness, myocardial extracellular matrix remodeling, heart concentric hypertrophy, and systemic inflammation. Validation of these in silico data was performed in vivo in patients with HFpEF by measuring the declining plasma concentrations of NOS2, the NLPR3 inflammasome, and TGF-β1 during 12 months of empagliflozin treatment. Using AI modelling, the main effect of empagliflozin in HFpEF is exerted via NHE1 and is focused on cardiomyocyte oxidative stress modulation supporting the potential use of empagliflozin in HFpEF [82], in accordance with observation of animal models discussed above.

Notwithstanding of the role of hyperacetylization, the calcium metabolism may still play a role as suggested from a recent study investigating mitochondrial and cytosolic calcium handling in intact cardiomyocytes from ZSF1-obese animals [83,84]. Mitochondrial and cytosolic calcium level at rest and during contraction are in the ZSF1-obese rat higher when compared with lean ZSF1-controls. Myocardial excitation-contraction coupling begins by membrane depolarization that triggers Ca^2+^ entry via L-type calcium channels, which in turn stimulate release of Ca^2+^ ions from the sarcoplsmaic reticulum with subsequent myofilament activation and contraction with sarcomere shortening [85]. To terminate Ca^2+^-mediated myofilament interaction, the amount of cytosolic Ca^2+^ must decline which requires inactivation of L-type calcium channels and sequestration of cytosolic Ca^2+^ by the sarcoplasmic reticulum Ca^−^ATPase (SERCA2a) back into the sarcoplasmic reticulum. Relaxation is a key player of HFpEF pathophysiology, therefore, it is not surprising that modifications in proteins that mediate Ca^2+^ homeostasis have been reported. ZSF1-obese rats show higher cytosolic Ca^2+^ level resulting from a disbalanced interaction between phospholamban (PLB) and sarcoplasmic endoplasmic reticulum calcium pump (SERCA2a). This disbalance reduces the reuptake of Ca^2+^ ions via the SERCA2a back into the sarcoplasmic reticulum. In ZSF1-obese rats, the SERCA2a protein content was unchanged between groups when compared to the lean controls, while PLB expression was higher in the ZSF1 obese rat. Furthermore, hypophosphorylated PLB has also an inhibitory effect on SERCA2a activity, suggesting that disbalance of the PLB/SERCA2a expression ratio together with hypophosphorylation increase cytosolic Ca^2+^ levels in diastolic dysfunction. Since isolated hearts from rats with metabolic syndrome and diabetes have shown incapacity to upregulate ATP synthesis and decreased free energy of ATP hydrolysis (ΔG∼ATP) at higher workloads, exercise will worsen diastolic relaxation in addition [86,87]. Additional evidence for a role of calcium homeostasis in diastolic dysfunction derives from a study conducted in an aged murine model recapitulating age-related cardiac changes compatible with HFpEF. These animals demonstrated an association between SERCA2 protein expression and age-dependent diastolic dysfunction [52]. Coherent with this, gene expression of SERCA2a was reduced in human cardiac allografts with diastolic dysfunction resulting in a relative increase of PLB gene expression [88], while regulatory subunits of the L-type calcium channel decrease their gene expression with diastolic dysfunction in the human cardiac allograft most likely as a compensatory mechanism aiming to decrease inflow of exogeneous Ca^2+^ ions via the central pore of the cardial L-type calcium channel [89]. Taken together, these results indicate that slowed calcium reuptake into the sarcoplasmic reticulum [90] may explain increased cytosolic and mitochondrial higher calcium concentration in the ZSF1-obese rats, but also in humans (Figure 1).

However, increased cytosolic Ca^2+^ is not only associated with altered excitation-contraction coupling, but also with increased mitochondrial Ca^2+^. The latter is a regulator of NADH (complex 1) activity, which may explain the observation that mitochondrial respiration and oxidative phosphorylation are reduced in HFpEF [91,92,93,94]. Furthermore, sustained increase of mitochondrial calcium accumulation may also turn detrimental because it may promote opening of mitochondrial permeability transition pore, ultimately leading to cellular apoptosis [95]. In summary, these results suggest that change of mitochondrial function can ultimately result in bottlenecks of metabolic flux, redox imbalance, protein modification, ROS-induced ROS generation, impaired calcium homeostasis, and local inflammation [96] with the potential to affect cardiomyocyte function [97].

### 2.3. Impact of Systemic Inflammation on Cardiomyocyte Function, Cardiac Fibrosis, and Vascular Function

Cardiac fibrosis is defined as a state in which excess deposition of collagen in the extracellular matrix occurs. In the resting state of a healthy heart, fibroblasts constantly modify the extracellular environment, while an overall disproportionately increased extracellular matrix volume is a hallmark of HFpEF [21]. The importance of this pathological increase of extracellular matrix is highlighted by the fact several profibrotic markers (chitinase 3-like protein 1 (a marker of hepatic fibrosis), FGF23, IL6, MMP7, and sST2) were among the eight serum biomarkers that most accurately predict prognosis in study participants of the TOPCAT trial [98]. In coherence, the Karolinska Rennes biomarker study investigating 87 biomarkers and 240 clinical markers for their association with New York Heart Association (NYHA) class and the combined endpoint of all-cause mortality and HF hospitalization in participants with LVEF > 45% sorted growth/differentiation factor-15, a member of the TGF ß superfamily, as the strongest predictor after adjusting for age, sex, and N-terminal proBNP [99].

However, fibrosis is mild to moderate in the majority of HFpEF patients, while only 26% suffer from severe fibrosis [100], questioning an exclusive role of the quantity of cardiac fibrosis [101]. This brings into claim that distribution and crosslinking of extracellular matrix proteins also affect mechanical properties of the myocardium suggesting these properties are relevant beyond quantity alone [102]. Diabetes [103] and obesity [104] may explain why the amount of cardiac fibrosis in not largely increased in the majority of HFpEF patients. Both comorbidities are associated with proinflammatory and oxidative stress (Figure 2C) [13] and, in addition, proinflammatory molecules are also implicated in the crosstalk between fibroblasts, cardiomyocytes, and vascular cells HFpEF [105]. In fact, myocardial collagen deposition in HFpEF is a consequence of the proliferation, differentiation of the fibroblast into a myofibroblast [106], and this differentiation is induced by tumor growth-factor ß (TGFß), a cytokine released by monocytes implicated in the vascular inflammatory response in HFpEF [106,107] (Figure 1). In addition, cardiomyocytes, fibroblasts, and vascular cells likewise synthesize and secrete proinflammatory cytokines such as IL-1, IL-6, and IL-8, colony-stimulating factors (CSF) of granulocyte or macrophage origin, and the chemotactic factor MCP1, which supports migration and extravasation of inflammatory cells into tissue [108]. Elevated IL-6 levels in macrophages [109] and cardiomyocytes [110] were shown to result in cardiac hypertrophy [111,112,113] and cardiac fibrosis via activation of IL-6 MAPK and IL-6-CaMKII-STAT3 pathways resulting in cardiac fibrosis and studies in rodents suggest that upregulated hypoxia-induced mitogenic factor (HIMF) in the heart may play a role [114].

Data from the longitudinal health-ABC study following an aged urban population provide further support for a role of sustained systemic inflammation and vascular endothelitis in development of HFpEF [115]. Increased plasma levels of Il-6 and tumor necrosis factor (TNF) α were in this longitudinal study predictive for incidence of HFpEF [115] and elevated IL-6 and TNFα levels in the human myocardium derange cardiomyocyte systolic and diastolic function (Figure 1) [116,117]. Indeed, the biomarker analysis in the TOPCAT study population sorted C-reactive protein, IL1β, IL-6, IL-10, immunoglobulin-like transcript 6, TNFα, TNFα receptor, and myeloperoxidase as the strongest predictors of prognosis in HFpEF [118]. These observations are compatible with reports that cardiovascular outcome in HFpEF [77,119,120,121,122,123,124,125] is associated with increased circulatory levels of interleukin-6 (IL-6) and TNFα as well as the soluble suppression of tumorigenicity 2 protein (ST2) and pentraxin plasma levels [126,127]. In HFpEF, increased blood levels of the latter have been associated with the comorbidities obesity, diabetes mellitus, chronic obstructive pulmonary disease, anemia, and chronic kidney disease [32,119]. This suggests that systemic inflammation is an important determinant for evolution towards HFpEF both in the clinical setting and in the murine models applying either a two-hit or three-hit strategy [73,74].

An increasingly popular idea about HFpEF is that systemic inflammation not only disrupts endothelial signaling, but also creates a stressful environment for the cardiomyocytes. In fact, an increased level of CD3^+^ cells and leucocytes positive for CD11a^+^, CD45^+^, and CD68nells was shown in right ventricular biopsy samples of HFpEF patients [107]. Further evidence for a role of immune cells in HFpEF derives from a study, which applied a broad T-cell inhibitor treatment in HFpEF patients resulting in slowed progression of cardiac hypertrophy and reduced activation and infiltration of T-cells and macrophages into the myocardium, and decreased loss of cardiomyocytes [128].

Last not least, systemic inflammation with the increased levels of pro-inflammatory mediators such as TNFα affects endothelial function of the coronary microvasculature with increased expression of endothelial adhesion molecules such as vascular cell adhesion molecule (VCAM) and E-selectin, as shown in myocardial biopsy samples from HFpEF patients [103,107,129]. Endothelial VCAM and E-selectin expression involve activation and the trans-subendothelial migration of circulating immune cells such as macrophages [107] into the myocardium due to vascular barriers weakening [100,107]. Macrophages infiltration has been demonstrated to be a key element of fibrosis through their interations with fibroblast leading to heart stiffness and myocardial dysfunction [130].

In parallel, endothelial cells are also affected by the systemic inflammation resulting in endothelial production of reactive oxygen species (ROS) due to activation of nicotinamide adenine dinucleotide phosphate oxidase [131]. This may also explain the high nitrosative/oxidative stress observed in HFpEF myocardium impacting on the mitochondrial function and thus the good working of cardiomyocytes [32,132,133]. Reduced NO bioavailability also promotes profibrotic action of the growth-promoting hormones endothelin-1, angiotensin II, and aldosterone favors collagen deposition [134], and more so in HFpEF, while less in HFrEF [135]. Furthermore, coronary microvascular endothelial inflammation leads to a reduced vasodilatation of the coronary microvascular bed as demonstrated by reduced acetylcholine-related vasodilation of the microcirculation in HFpEF [136]. This reduced response can explain left ventricular (LV) diastolic dysfunction [136] and a deficient systemic vasodilator response could also explain the reduced exercise tolerance [137]. However, this loss of exercise tolerance can be compensated thanks to exercise training program [138,139], which also to some extent improves diastolic LV dysfunction [138,139]. Indeed, there is evidence that either improvement is related to upregulation of endothelial nitric oxide synthase (eNOS) [138,139].

## 3. Conclusions

The expanding prevalence of HFpEF throughout the world and the lack of truly effective therapeutic options clearly ask for a focus on this syndrome. Current evidence suggests that HFpEF syndrome is at least partially a systemic syndrome, and there is clearly a need to broaden the focus of research and to create animal models incorporating the complex interactions between different comorbidities and organ systems. Although small rodents are undeniably the premier model system for discovering new mechanisms and targets, the major push is in-depth characterization of HFpEF patients using cutting-edge imaging based on latest cardiovascular magnetic resonance (CMR) [140,141] conjoint with genomic, proteomic, and metabolomic analysis. However, greater procurement of samples of heart, skeletal muscle, fat, or other tissues from patients with HFpEF is required similar to what is commonly performed in oncology. Machine learning will be useful for this analysis and may in humans and in animal models help to better identify potential therapeutic targets. In the meantime, the experimental models, which more lately recapitulate the clinical situation, move in the right direction with improved capture of the multidimensionality of the HFpEF syndrome.

## Figures and Tables

**Figure 1 cells-11-00867-f001:**
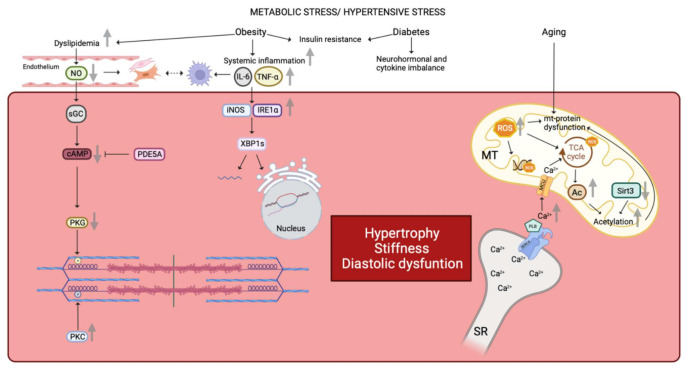
Similarities in human and animal HFpEF pathophysiology. Metabolic stress and hypertensive stress impaired cardiomyocyte functionality. Obesity favors dyslipidemia, which leads to NO reduction and finally a reduced PKG activity by the NO-sGC-cAMP pathway. Reduced PKG activity is correlated with a decreased N2B phosphorylation. On the contrary, PKC activity is increased in HFpEF disease leading to an increased phosphorylation state of the titin’s PEVK domain. Moreover, reduced NO bioavailability promotes fibroblasts and myofibroblasts proliferation. These latter interact with infiltrated macrophages which are linked with the inflammatory state. Indeed, obesity favors systemic inflammation with an increased concentrations of IL-6 and TNF-α. Increased plasma levels of Il-6 and TNF-α impair gene expression as well as the correct folding of proteins by the iNOS-IRE1α-XBP1 pathway. Both obesity and diabetes comorbidities are linked with insulin resistance, neurohormonal and cytokine imbalance affecting the cardiomyocyte environment. Finally, aging alters mitochondrial function by increased ROS production. ROS impact on mitochondrial DNA, mitochondrial proteins, and TCA cycle. Impairment of TCA cycle is also due to the increased mitochondrial Ca^2+^ concentration. Impairment of the TCA cycle leads to an increased Acetyl-CoA concentration, due to a decreased expression of mitochondrial deacetylase Sirt3, and results in hyperacetylation of mitochondrial proteins. Furthermore, increased mitochondrial Ca^2+^ concentration results in increased PLB activity, which inhibits SERCA and promotes cytosolic Ca^2+^ accumulation, which passes into the mitochondria via the MCU. Altogether, these pathways lead to a deterioration of the cardiomyocyte state by promoting hypertrophy, stiffness, and diastolic dysfunction. NO: nitric oxide, PKG: protein kinase G, sGC: soluble guanylate cyclase, cAMP: cyclic adenosine monophosphate, PKC: protein kinase C, PEVK domain: proline-glutamate-valine-lysine domain, Il-6: interleukine 6, TNF-α: tumor necrosis factor α, iNOS: inducible nitric oxide synthase, IRE1α: inositol requiring endoribonuclease 1α, XBP1: X-box binding protein 1, ROS: reactive oxygen species, TCA: tricarboxylic acid, Ca^2+^: calcium, SERCA: sarcoplasmic/endoplasmic reticulum calcium ATPase, PLB: phospholamban, MCU: mitochondrial calcium uniporter, Ac: acetyl group, Sirt3: Sirtuin 3, SR: sarcoplasmic reticulum, MT: mitochondria, mt: mitochondrial, PDE5A: phosphodiesterase type 5A, P: group phosphate.

**Figure 2 cells-11-00867-f002:**
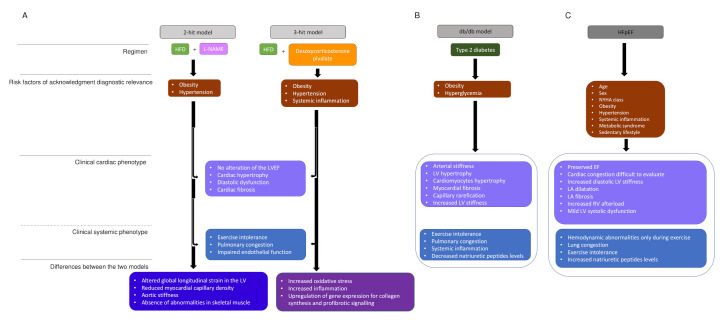
HFpEF models and human HFpEF diagrams. (**A**). Two-hit model versus three-hit model. two-hit model is based on exposition of mice to HFD and L-NAME to induce metabolic stress (obesity) and mechanical stress (hypertension). Obesity and hypertension are risk factors that underlie HFpEF. Three-hit model is induced by the treatment of mice with HFD and desoxycorticosterone pivalate, which lead to the same phenotypes as in the two-hit model and in addition present a systemic inflammation. The central boxes include the clinical cardiac phenotype and the clinical systemic phenotype present in both strategies. The two last boxes represents the differences observed in, respectively, two-hit and three-hit models. (**B**). The db/db model summarizes the phenotype observed in a murine model, which presents type 2 diabetes. (**C**). The human HFpEF diagram summarizes the clinical phenotype observed and HFpEF singularities. HDF: high fat diet, L-NAME: N-nitro-L-arginine methyl ester, LV: LV: left ventricule, LVEF: left ventricular ejection fraction, NYHA: New York heart association, EF: ejection fraction, LA: left atrial.

## Data Availability

Not applicable.

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
