# Peer review of "The Cardiomyocyte in Heart Failure with Preserved Ejection Fraction—Victim of Its Environment?"

_cells, 2022, doi:10.3390/cells11050867_

Round 1

Reviewer 1 Report

All previous suggestions have been adequately addressed. No further comments.

Reviewer 2 Report

This review is an interesting topic and authors made relevant figures.

This manuscript is a resubmission of an earlier submission. The following is a list of the peer review reports and author responses from that submission.

Round 1

Reviewer 1 Report

Rocca et al., review findings from clinical studies and pre-clinical animal models to highlight the complexities associated with HFpEF. To highlight potential therapeutic avenues, the authors discuss the impact of cardiomyocytes, comorbidities (age, hypertension, obesity, diabetes), fibrosis, and system inflammation in mediating HFpEF onset and progression. A comprehensive understanding of the molecular mechanisms associated with HFpEF may facilitate the discovery of novel treatment targets for improving clinical outcomes of HFpEF patients.  

Major comments:

  1. Several reviews have already discussed the findings presented in this manuscript, and so, this reviewer is uncertain as to what novel perspective the authors aim to offer.
  2. The title is misleading, as it seems to suggest an evaluation of studies that have investigated cardiomyocyte-specific perturbations that are associated with HFpEF. However, there’s only a small section regarding alterations in cardiomyocyte structure/function, with a majority of content reflecting the impact of other factors, such as comorbidities, fibrosis, and inflammation.   
  3. Most of the studies reviewed here investigated disease mechanisms in whole hearts. If the authors intend to make this review cardiomyocyte focused, further discussion on how these mechanisms could potentially impede cardiomyocyte function is required. E.g., line 292; …URP affect the cardiomyocytes… how does this affect cardiomyocytes directly?   
  4. The abstract is not very clear, and does not support the title.

Minor comments:

The spelling and grammar should be checked thoroughly. E.g., line 17; diabetes is repeated, line 179; …due increased…; line 253; Inhibition improved – what is inhibited?; line 287; S-nirosylation

Reviewer 2 Report

This review merges observation from animal models and human HFpEF disease with the intention to converge pathophysiological aspects and clinical knowledge.

This referee found the article well-prepared without major suggestions of modification. As a minor concern I found the language very hermetic as I needed to read more than once the same sentence to understand the meaning. A more friendly language would be more interesting for fluency.

Reviewer 3 Report

The review by Rocca et al. focuses on in-vitro changes in HFpEF that are mediated by differential triggers. The authors nicely summarize cardiomyocyte related changes during HFpEF and elucidate the impact of comorbidities, fibrosis and inflammation on HFpEF genesis in different models.  

Major:

-The authors partly focus on changes related to the 2-Hit and 3-Hit model (and put emphasis on them in the abstract), yet these models are not even mentioned in other paragraphs (e.g. in the structural and functional changes of the cardiomyocyte in hfpef paragraph), structure-wise it might be advantagous to discuss each contributing factor (cardiomyocyte, comorbidities, fibrosis, inflammation etc) in the context of the same models/human.

-Implications on mitochondrial calcium are not discussed very well (see e.g. https://pubmed.ncbi.nlm.nih.gov/31520455/)

-The role of cytosolic Calcium removal for diastolic dysfunction and altered excitation-contraction coupling in general could be discussed in the cardiomyocyte paragraph.

-Since the (somewhat controversal) results of EMPEROR-Preserved are quite intriguing it would be nice to add a paragraph considering evidence for subcellular effects of SGLTi in HFpEF.

Minor:

-Abstract (ln 15): "almost normal" or "almost normal and normal"

-ln 83 "only a small portion of patients was diagnosed in common.", consider revising this sentence.

-ln 84: consider emphasising "LVEDP at rest". Maybe this subset of patients was merely hypertensive?! What other markers of HFpEF were found in the study

-ln 161: "with different risk", consider revising this sentence.